# Algorithm Validation for Quantifying ActiGraph™ Physical Activity Metrics in Individuals with Chronic Low Back Pain and Healthy Controls

**DOI:** 10.3390/s24165323

**Published:** 2024-08-17

**Authors:** Jordan F. Hoydick, Marit E. Johnson, Harold A. Cook, Zakiy F. Alfikri, John M. Jakicic, Sara R. Piva, April J. Chambers, Kevin M. Bell

**Affiliations:** 1Department of Bioengineering, University of Pittsburgh, Pittsburgh, PA 15213, USAzakiy.alfikri@pitt.edu (Z.F.A.); ajchambers@pitt.edu (A.J.C.); 2Department of Orthopaedic Surgery, University of Pittsburgh, Pittsburgh, PA 15213, USA; 3Department of Internal Medicine, University of Kansas Medical Center, Kansas City, KS 66160, USA; 4Department of Physical Therapy, University of Pittsburgh, Pittsburgh, PA 15213, USA; spiva@pitt.edu; 5Department of Health and Human Development, University of Pittsburgh, Pittsburgh, PA 15213, USA

**Keywords:** actigraphy, chronic low back pain, ActiGraph, inertial measurement units, algorithms

## Abstract

Assessing physical activity is important in the treatment of chronic conditions, including chronic low back pain (cLBP). ActiGraph™, a widely used physical activity monitor, collects raw acceleration data, and processes these data through proprietary algorithms to produce physical activity measures. The purpose of this study was to replicate ActiGraph™ algorithms in MATLAB and test the validity of this method with both healthy controls and participants with cLBP. MATLAB code was developed to replicate ActiGraph™’s activity counts and step counts algorithms, to sum the activity counts into counts per minute (CPM), and categorize each minute into activity intensity cut points. A free-living validation was performed where 24 individuals, 12 cLBP and 12 healthy, wore an ActiGraph™ GT9X on their non-dominant hip for up to seven days. The raw acceleration data were processed in both ActiLife™ (v6), ActiGraph™’s data analysis software platform, and through MATLAB (2022a). Percent errors between methods for all 24 participants, as well as separated by cLBP and healthy, were all less than 2%. ActiGraph™ algorithms were replicated and validated for both populations, based on minimal error differences between ActiLife™ and MATLAB, allowing researchers to analyze data from any accelerometer in a manner comparable to ActiLife™.

## 1. Introduction

Low back pain (LBP) is the leading cause of disability worldwide, affecting over 500 million people globally [1,2,3]. It is estimated that up to 80% of adults will experience an episode of back pain in their lives [4]. Physical activity has a beneficial effect on musculoskeletal conditions, including LBP [5,6,7]. However, the appropriate amount of recommended physical activity in the LBP population is questioned, as it has been shown that both too much and too little activity may be associated with LBP [8,9]. Unfortunately, some LBP cases do not resolve, leading to chronic low back pain (cLBP), which is defined as pain between the inferior border of the ribcage and gluteal fold lasting more than 12 weeks, resulting in pain on at least half the days in the past 6 months [10].

Research suggests that engaging in regular physical activity can reduce the risk of developing cLBP, as well as improve symptoms and function in those who already experience LBP [11,12]. More specifically, core strengthening, or spinal stabilization exercises, can help strengthen the muscles supporting the lower back and improve flexibility and range of motion (ROM), which can contribute to a healthier spine [13,14]. However, high-impact activities, such as running or jumping, may put excessive stress on the spine and contribute to pain or injury. Similarly, activities that involve repetitive bending, twisting, or lifting may also be problematic for those with LBP [9]. Conversely, a meta-analysis published in 2019 that examined the association between physical activity and LBP found an inverse relationship between physical activity and LBP [8,15]. A significant limitation of the meta-analysis is that most of the included studies used self-administered questionnaires that likely produce overestimation and recall bias, highlighting the importance of quantitative physical activity assessment.

In recent years, there has been an increasing emphasis on wearable technologies and tools to measure actigraphy and other physical activity metrics. Common amongst numerous actigraphy devices is that they contain accelerometers or inertial measurement units (IMUs), which are devices that measure and report acceleration, along with other measures, including angular rates, magnetic fields, which can be used to estimate orientation. With all these options available, it is often difficult to know which devices or options are best for a given application.

Actigraphy is a non-invasive technique for measuring free-living physical activity with devices that collect accelerometry data for movement detection and positional changes. Actigraphy devices are traditionally worn on the wrist or waist. The use of actigraphy to objectively measure physical activity has been validated in various studies providing an alternative to self-reported measures alone [16,17]. The collected raw accelerometry data are quantified into physical activity measures, including activity counts and step counts [18]. Activity counts can be summed over an epoch, or time frame, and categorized into cut points to represent the intensity of activity over the epoch. Freedson Adult (1998) cut points are one commonly used method, which were developed to correspond to common metabolic equivalent of task (MET) categories [19]. A commonly used epoch length is 60 s, resulting in counts per minute (CPM), that can then be categorized into a cut point to describe the activity intensity during each minute of data collected.

ActiGraph™ (ActiGraph™, Inc., Pensacola, FL, USA) accelerometers are widely used to assess physical activity in research settings, with over 20,000 papers published using these devices [18]. ActiGraph™ sensors are also the most commonly used sensors for assessing physical activity in individuals with cLBP [20]. The software, ActiLife™, processes raw acceleration data through proprietary algorithms to produce physical activity measures such as activity counts, counts per minute (CPM), step counts, and intensity cut points. Many studies assessing physical activity collect data from an ActiGraph™ worn on the waist attached to the body by a belt, which is typically removed while sleeping [21]. 

Although ActiGraph™ sensors are the most commonly utilized sensors for assessing physical activity in individuals with cLBP, greater than 60% of studies in a recent scoping review used alternative devices [20]. The high variability in the model of the devices, the algorithms for analyses, and wear location makes it difficult to directly compare findings and limits progress in this important research area. Moreover, the proprietary nature of the data analyses algorithms limits opportunities to customize and optimize the outcomes for the application and population of interest [22]. For example, for individuals with cLBP, placing IMUs along the lumbar region may reveal new insight into physical activity or spinal movements. Moreover, the flexibility to adjust the cut points and/or data analysis algorithms to output additional kinematic analysis metrics such as velocity or ROM may reveal new insights and/or treatment opportunities. 

To translate the data collected through a custom IMU, the ActiGraph™ proprietary algorithms must be reproduced and validated to obtain comparable results. Until recently, the ActiGraph™ activity counts algorithm was completely proprietary, making it difficult for researchers and clinicians to translate accelerometry data to activity counts in a standardized way unless the data were collected and analyzed through an ActiGraph™ device. Therefore, the purpose of this work was to replicate ActiGraph™ algorithms in MATLAB (2022a) and to test the validity of the algorithms to analyze ActiGraph™ data and reproduce the outputs of the ActiLife™ (v6) software for both healthy controls and the cLBP population. It is hypothesized that the custom MATLAB algorithms can reproduce the outputs of the ActiLife™ software when analyzing ActiGraph™ collected data with a percentage error of less than 5% for all outcomes. The outcomes selected for replication and validation were activity counts, step counts, and activity intensity level (minutes in sedentary, light, moderate, vigorous, and very vigorous activity) as they are the most commonly reported outcomes for assessing physical activity in individuals with cLBP [20].

## 2. Materials and Methods

### 2.1. Algorithm Development

Figure 1 displays the proposed algorithm workflow that is described in detail throughout this section. To summarize, the raw *y*-axis acceleration data are calculated into step counts and activity counts. The activity counts are summed into CPM and then categorized based on the Freedson Adult (1998) cut points [19]. 

In 2022, ActiGraph™ published a description of their proprietary activity count algorithm used to process raw acceleration data in ActiLife™, which served as the foundation for the custom MATLAB algorithms developed for this project [23]. The MATLAB algorithms utilized a sampling frequency of 60 Hz and an epoch length of 60 s [18].

The seven processing steps described by Neishabouri, et al. [18] were programmed into a MATLAB function. The steps are down-sampling the signal to 30 Hz, band-pass filtering the down-sampled signal, rescaling the filtered signal by multiplying it by a factor of 17.127404, taking the absolute values of the rescaled signal, applying a threshold to the signal so that all values greater than 128 are set to 128 and all values between 0 and 4 are set to 0, further down-sampling it to 10 Hz, and summing the down-sampled signal within a 60 s epoch length. The inputs to the function are the raw acceleration data, sampling frequency, and epoch length. The output is the CPM vector for all data collected. The total activity counts are calculated by summing the entire vector. Although most actigraphy downstream analysis focuses on the activity counts in the vertical (*y*-axis), the activity counts in the *x*-axis and *z*-axis were also calculated. 

When evaluating physical activity over extended periods of time, a wear time algorithm must be used to detect periods when the person takes the device off. ActiGraph™ provides two options in ActiLife™: Troiano (2007) [24] and Choi (2011) [25]. The Troiano method was chosen for this application, as the Choi method was validated in a smaller sample size, a majority of which were children [24,25,26]. As described by Troiano, the wear time algorithm detects non-wear time from 60 s *y*-axis epoch counts [24]. Non-wear time was defined by an interval of at least 60 consecutive minutes of zero activity counts, with allowance for 1–2 min of counts between 0 and 100 [24]. The function input is the CPM vector calculated through the activity counts algorithm, and the output is a non-wear vector consisting of 1’s and 0’s, where “1” is defined as wear time during the minute, and “0” is defined as non-wear time. The algorithm removes CPM during non-wear time, resulting in only wear time. The new CPM vector is summed to achieve the total counts, excluding non-wear time. The total wear time in minutes is determined by the length of the vector. The resulting variables are the CPM excluding non-wear time in the X, Y, and Z directions, totals in all axes, and total wear time in minutes. All further analysis is based on the CPM excluding non-wear time. 

Since the *y*-axis is in-line with gravity, it is the most common axis analyzed when calculating activity counts. However, the ActiGraph provides a 3-axes accelerometer data; therefore, it is also possible to consider all three axes. The vector magnitude, which reflects all three axes, was calculated according to Equation (1).
(1)$$\mathrm{Vector}\;\mathrm{Magnitude}\;\mathrm{Counts}\;\mathrm{Total}=\sqrt{{\left(X\;Axis\;Total\;Counts\right)}^{2}+{\left(Y\;Axis\;Total\;Counts\right)}^{2}+{Z\;Axis\;Total\;Counts}^{2}}$$

Using the CPM output from the activity counts algorithm and wear time algorithm, the code assigns each CPM to the corresponding Freedson cut point, as described in Figure 1. The total minutes in each cut point are summed to calculate the total minutes spent in sedentary, light, moderate, vigorous, and very vigorous activity. The percentages spent in each cut point are calculated by dividing the total minutes spent in a cut point by the total wear minutes multiplied by 100.

Another common activity metric is step counts. Although many step count algorithms exist, for the sake of consistency, the step count algorithm implemented in ActiLife™ was modified for this application. Upon request, ActiGraph™ customer support provided a draft version of their step count algorithm, and a step count algorithm was developed in MATLAB based on the description and pseudo-code they provided [27,28]. The algorithm includes down-sampling the signal to 30 Hz, band-pass filtering the down-sampled signal, rescaling the filtered signal by multiplying it by a factor of 17.127404, applying threshold to identify step as a crossing below zero into −4 followed by a crossing above zero into 4, and summing the number of steps identified. Because this step count algorithm is not published or validated, pilot testing was used to verify results as the algorithm was developed. 

### 2.2. Participants

This study consisted of participants with cLBP and healthy controls. All subjects gave their informed consent before they participated in this study. This study was conducted in accordance with the Declaration of Helsinki, and the protocol was approved by the Ethics Committee of The University of Pittsburgh (STUDY20020091). Potential participants were recruited using the Pitt + Me^®^ recruitment registry (Clinical and Translational Science Institute (CTSI), University of Pittsburgh, Pittsburgh, PA) and all were phone-screened to ensure they met inclusion/exclusion criteria. Healthy participants were included if they had no history of LBP within the last two years, no current low back symptoms, and no back surgeries. Participants with cLBP were included if they had a history of LBP greater than twelve weeks with persisting symptoms ≥ 50% of the time [10]. Recruitment occurred between 1 June 2021 and 17 December 2021. Participants with cLBP were matched to asymptomatic controls who were as close in age as possible. Asymptomatic participants whose ages fell in those decades were screened until the number of asymptomatic participants was equal or nearly equal to the participants with cLBP. The goal was to have the same number of participants with cLBP and without, matching gender as much as possible.

### 2.3. Free-Living Validation

An ActiGraph™ GT9X (ActiGraph, Pensacola, FL) was used in this work, a tri-axial accelerometer with the following coordinate system: *y*-axis = vertical direction, *x*-axis = horizontal direction, and *z*-axis = perpendicular direction when oriented as displayed in Figure 2. Participants were instructed to wear an ActiGraph™ GT9X for up to seven days on their non-dominant hip attached by a belt, as shown in Figure 3, except while sleeping or showering. Seven days was selected for this study as it is the most common duration utilized for assessing physical activity in cLBP based on a scoping review [20].

### 2.4. Data Processing and Analysis

Software needed to process the data included ActiSync™ (ActiGraph™, Pensacola, FL, USA), CentrePoint (ActiGraph™, Pensacola, FL, USA), ActiLife™ (ActiGraph™, Pensacola, FL, USA), and MATLAB (2022a). ActiSync™ software was used to upload ActiGraph™ raw data to CentrePoint, ActiGraph™’s data management software. Raw data were exported through CentrePoint. The raw data for all participants were processed in both ActiLife™ and MATLAB, and the results were compared. Mean results for total *y*-axis activity counts, total vector magnitude counts, total step counts, and total minutes spent in each category based on the Freedson cut points were calculated [19]. For analysis, percentage errors between the two methods were calculated for all participants and separated between participants with cLBP and healthy controls to test the algorithm performance in both populations.

## 3. Results

### 3.1. Participant Demographics

Twenty-four participants, twelve cLBP and twelve healthy controls, were recruited and consented to participate in this study. Participant demographics are displayed in Table 1. The ages of participants ranged between 21 and 69 years old. 

### 3.2. Free-Living Validation

Mean results for total *y*-axis counts, total vector magnitude counts, total step counts, and total minutes spent in each Freedson cut point in ActiLife™ and the developed MATLAB algorithms are displayed in Table 2, as well as percent errors between the two methods. Total percentage errors for all 24 participants, as well as separated by group, were all less than 2%. The lowest percent error was the activity counts algorithm with an error of 0.00% for all participants, 0.00% for the healthy group, and 0.01% for the cLBP group. The highest percent error was for the total minutes in the sedentary category, with an error of 1.21% for all participants, 1.31% for the healthy group, and 1.11% for the cLBP group.

## 4. Discussion

The purpose of this work was to replicate ActiGraph™ algorithms in MATLAB (2022a) and to test the validity of these algorithms to analyze ActiGraph™ data and reproduce the outputs of the ActiLife™ software for both healthy controls and the cLBP population. It was hypothesized that the custom MATLAB algorithms can reproduce the outputs of the ActiLife™ software when analyzing ActiGraph™ collected data with a percent error of less than 5% for all outcomes. This work was successful in developing algorithms in MATLAB replicating techniques used in ActiLife™ to quantify physical activity metrics of wearable sensor accelerometry data including activity counts, CPM, cut points, wear time, and step counts. Percent errors for all 24 participants, as well as separated by group, were all less than 2%, which was well below the hypothesized target of <5%. 

Several studies have attempted to reverse engineer the algorithm with similar goals of standardizing the method to quantify activity counts with custom devices. Peach, et al. [28] attempted to determine the form of the filter used in the ActiLife™ software, which is the first step in processing raw acceleration data into counts, and tested the developed filter on various frequencies. They concluded that the filtration method can drastically alter the results, and stressed the need for a published, standardized method [28]. In another study, Rao, et al. [29] attempted to go beyond just the filtering step and explored the additional processing steps needed to obtain the final desired end products, which are activity counts. Rao, et al. [29] were able to design a method that closely matches ActiLife™, but the method was never fully validated. Brønd, et al. [22] described a method in MATLAB to quantify activity counts from raw acceleration data using a custom device and tested the validity of the method in both a mechanical validation and a 24 h free-living experiment. Brønd, et al.’s [22] validation resulted in a relative difference ranging from 0.5% to 4.7% with a group mean of 2.2%. In contrast, the relative difference during this study’s seven-day free-living validation resulted in a relative difference group mean of 0.00% for activity counts, which was an improvement over Brønd, et al. [22]. To our knowledge, this current paper is one of the most comprehensive studies published to date that has attempted to replicate the outputs of the ActiLife™ software having included activity counts, non-wear time, CPM, cut points, and step counts in a single application, emphasizing the unmet need and novel methods of this work.

These validated algorithms allow researchers and clinicians to calculate physical activity measures equivalent to those collected through an ActiGraph™. These algorithms can be a baseline for researchers interested in analyzing physical activity with a device that collects accelerometry data. The algorithms may have to be modified for the specific device, but they can assist in overcoming the beginning challenges. For example, the device’s sampling frequency and coordinate system would have to be known and modified in the code. Additionally, the location of the device on the body should be close to the person’s center of gravity, as these algorithms would not be appropriate for a wrist- or ankle-worn device where the accelerations would be vastly different. This work validated the algorithms in participants with cLBP but could be extended to any other population including other chronic conditions where physical activity is crucial to treatment. Additionally, it could be extended to other populations including athletes or the military, where assessing physical activity could help optimize performance.

## 5. Conclusions

The purpose of this work was to replicate ActiGraph™ algorithms in MATLAB (2022a) and to test the validity of these algorithms to analyze ActiGraph™ data and reproduce the outputs of the ActiLife™ software for both healthy controls and the cLBP population. All metrics evaluated, including activity counts, CPM, cut points, wear time, and step counts, resulted in a percentage error of less than 2% for both asymptomatic and cLBP participants. This work sets the stage for researchers to use MATLAB to assess accelerometry data in a comparable manner to ActiLife™, while allowing for more transparency and more accessible iterative improvements and customization for specific populations, like cLBP assessment. 

Limitations to this study include validating the algorithms on only 24 participants, 12 with cLBP and 12 healthy controls, and not testing the algorithms on data collected through a custom device. Additionally, the majority of the participants were female (N = 18), which may limit generalizability of the findings due to potential gender differences in cLBP and physical activity levels. Although this study demonstrated the validity of the custom MATLB algorithms, further work could focus on the small differences between methods, which may further improve results. For example, the counts algorithm developed in MATLAB uses a built-in “resample” command, which could be different from the one used in ActiLife™. 

The next steps for this study include adapting the validated algorithms and applying them to custom brand IMUs mounted on the spine in participants with cLBP and healthy controls. Data collected through an ActiGraph™ worn on the waist will be compared to data collected through the custom brand IMUs worn on the spine to interpret differences in device and location and to determine the feasibility of this method in participants with cLBP. Moreover, future work will aim to use the activity data to identify not just the volume of activity but possibly the type of activity, which may be very important for patients with cLBP. 

## Figures and Tables

**Figure 1 sensors-24-05323-f001:**
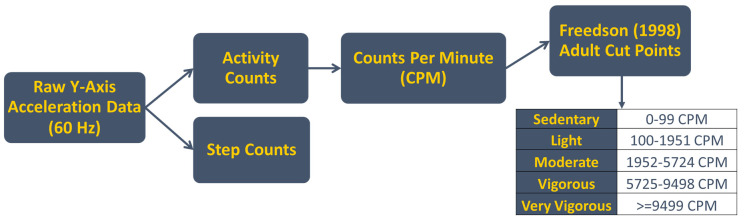
Algorithm workflow. Raw *y*-axis data are calculated into activity counts and step counts based on the accelerometer signals, then are summed into counts per minute (CPM) with Freedson Adult cut points for categorization. These data are then summarized by the total time and percentage of overall time spent in each categorization. The coordinate system for the ActiGraph™ is shown in Figure 2.

**Figure 2 sensors-24-05323-f002:**
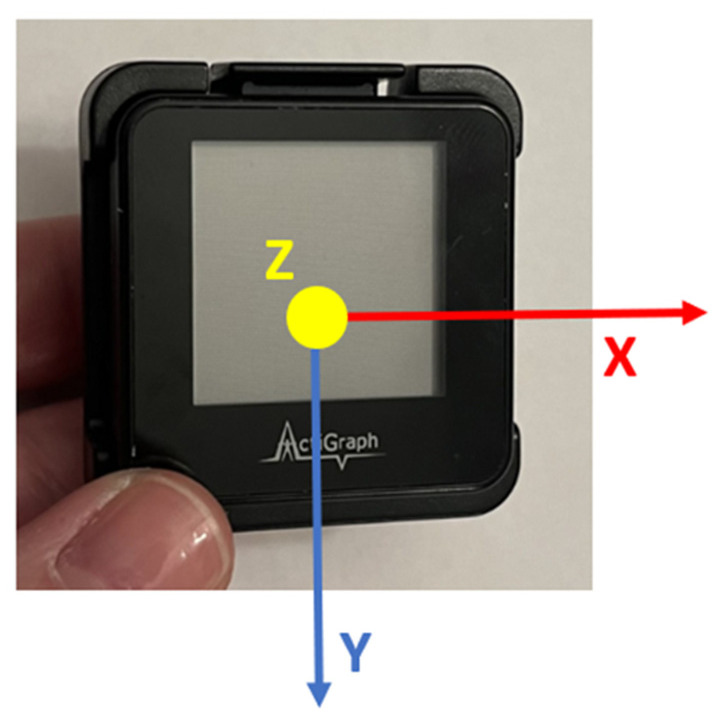
ActiGraph™ GT9X Coordinate System where X = horizontal, Y = vertical, Z = perpendicular.

**Figure 3 sensors-24-05323-f003:**
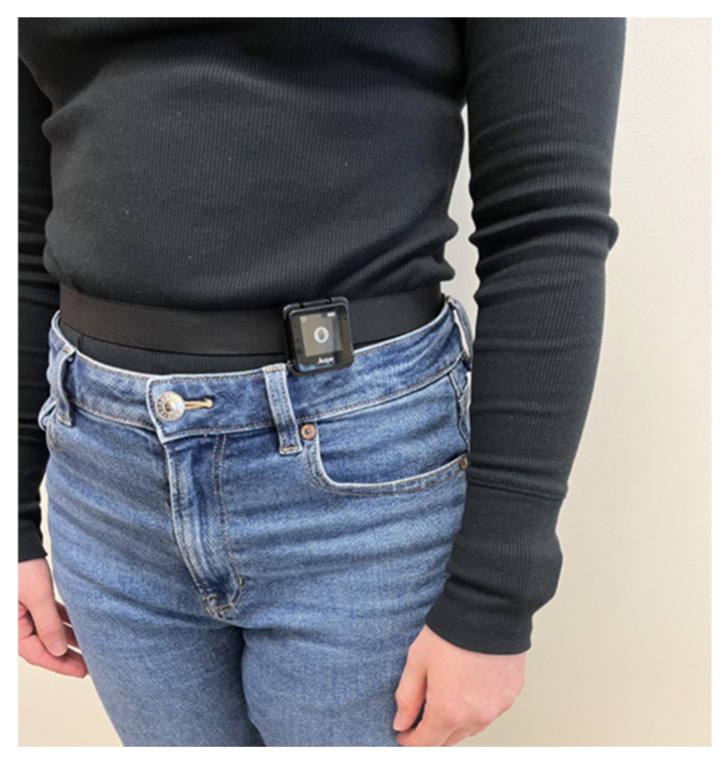
ActiGraph™ GT9X secured to the waist with a belt.

**Table 1 sensors-24-05323-t001:** Participant demographics.

Condition	Gender	Average Age (Years ± SD)
cLBP	Females: N = 10 Males: N = 2	41.9 ± 16.8
Healthy	Females: N = 8 Males: N = 4	42.4 ± 16.5
Total	Females: N = 18 Males: N = 6	42.2 ± 16.3

cLBP = chronic low back pain, SD = standard deviation.

**Table 2 sensors-24-05323-t002:** Results of the seven-day free-living validation including total error (N = 24), and error separated by participants with chronic low back pain (cLBP) (N = 12) and healthy controls (N = 12).

	ActiLife™	MATLAB	Error (%)	Healthy Error (%)	cLBP Error (%)
Total *y*-Axis Activity Counts	1,849,697.0	1,849,615.5	0.00	0.00	0.01
Total Vector Magnitude Counts	3,486,535.2	3,494,540.1	0.23	0.31	0.15
Total Step Counts	49,850.3	49,884.9	0.07	0.09	0.05
Total Minutes in Sedentary	3573.4	3616.8	1.21	1.31	1.11
Total Minutes in Light	1629.8	1629.8	0.00	0.01	0.01
Total Minutes in Moderate	225.8	225.4	0.18	0.08	0.40
Total Minutes in Vigorous	17.0	17.0	0.24	0.56	0.00
Total Minutes in Very Vigorous	0.2	0.2	0.00	0.00	0.00

cLBP = chronic low back pain.

## Data Availability

The generated datasets from this study are available upon request to the corresponding author.

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
