# Peer review of "Algorithm Validation for Quantifying ActiGraph™ Physical Activity Metrics in Individuals with Chronic Low Back Pain and Healthy Controls"

_sensors, 2024, doi:10.3390/s24165323_

Round 1

Reviewer 1 Report

Comments and Suggestions for Authors

The study's results are presented clearly and concisely. Furthermore, it addresses a relevant and valuable topic in the realm of physical activity monitoring, particularly for people with chronic low back pain (cLBP), and covers some physical activity metrics, such as activity counts, step counts, and categorization into physical activity intensity cut-points.

However, some aspects of the manuscript could be improved, and the study has some limitations, such as a small sample size (24 individuals, 12 with low back pain and 12 healthy controls), a relatively short validation period (seven days), and the fact that the majority (75%) of participants were female, which limits the generalizability of the findings due to potential gender differences in cLBP and physical activity levels.

The low percentage of errors observed (less than 2%) indicates that the replication of ActiGraph algorithms in MATLAB was done quite successfully. Nonetheless, the manuscript would benefit from a more detailed comparison and discussion of the ActiGraph algorithms and their MATLAB replications, taking into account that the accuracy of the ActiGraph GT9X is influenced by a number of factors, including measurement conditions, walking speed, device positioning, and the filtration method used, with previous studies indicating that there is a tendency to underestimate the number of steps in real-world situations. The manuscript also does not discuss energy expenditure or the validity of ActiGraph's estimation, nor does it compare and independently validate the results with other physical activity and energy expenditure measurement methods.

In addition to taking into account the previous comments, there are some improvements that can be made to the manuscript structure and writing quality. First, the Discussion's final two paragraphs should be moved to the Conclusions section, where they make more sense. Citations must be made in the Discussion section immediately after the authors' names (Rao et al. [28], Bond et al. [21]), not at the end of each phase. Finally, the References must be consistent, including the proper use of uppercase and lowercase initials and the use of ISO-4 abbreviations for the names of periodicals. References [22] and [26] may be clearly identified and completed.

Comments on the Quality of English Language

The manuscript is reasonably well-written and no more serious errors were detected; however, careful reading of the manuscript is recommended to improve the writing of some sentences.

Reviewer 2 Report

Comments and Suggestions for Authors

The authors present the results of their work replicating ActiGraph algorithms in MATLAB (2022a) and the validity of these algorithms for analyzing ActiGraph data and reproducing ActiLife software outputs.

However, the work has important weaknesses:

- the authors must explain the motivation for this work, the need to replicate algorithms already implemented;

- no details are provided in the article regarding the algorithms implemented, while these details are necessary to understand the work carried out;

- from Figure 1 it seems that only the data acquired along the Y axis are processed, while in the explanation provided in the text (page 3, rows 110-115) the data obtained on all three axes are taken into consideration (see equation 1);

- in Figure 1 the categorization of the activity level is implemented in absolute terms, while in the description provided in the text the authors speak of percentage values. Authors must therefore categorize activity levels into percentage values;

- table 2 cannot be divided into two pages.

Reviewer 3 Report

Comments and Suggestions for Authors

In this manuscript, the authors report on a study to replicate ActiGraph™ algorithms in MATLAB (2022a) and to test the validity of these algorithms to analyze ActiGraph™ data and reproduce the outputs of the ActiLife™ software for both healthy controls and the cLBP population.

Despite that the results reveal the validity of the custom MATLAB algorithms can reproduce the outputs of the ActiLife™ software when analyzing ActiGraph™ collected data with a percent error of less than 5% for all outcomes, it is still rather confusing for me that why this study is necessary or important to replace the commercially available ActiLife™. Does it make any sense of improving the activity ability of the cLBP patients?  Any novelty for this study by algorithm to quantify the physical activity with wearable sensor accelerometry? Moreover, the activity counts, cut points, wear time, and step counts seem not quite clearly relevant to the cLBP behavior itself in comparison to the healthy counterparts.

Based on the above consideration, I am afraid the manuscript does not meet the requirement of journal of sensors in current form. I recommend resubmission after major revision with more solid criteria and explanation.

Round 2

Reviewer 2 Report

Comments and Suggestions for Authors

The authors have met the requirements of the previous review, and the article can be published in its current form.

Reviewer 3 Report

Comments and Suggestions for Authors

The authors have addressed the questions and showed explanation more clearly in this revised manuscript, especially about the aim and method of the study on the assessment with the home-made algorithm compared to the commercially available test software.

In total, I think it can be accepted for publication in Sensors upon addressing some minor questions as follows.

1. It would be better to show some comparable data, for instance, some segments of recorded data of ActiGraph™ sensors and the outcome figures or charts of the Matlab code.

2. In Table 3, there are many results related to the time and counts, what else results in quantification could be provided then?

3. Some typying and syntax errors can be found in the text, please correct them.

Comments on the Quality of English Language

English is fine except for some syntax error and typying error.
